# Trend of HPV Molecular Epidemiology in the Post-Vaccine Era: A 10-Year Study

**DOI:** 10.3390/v15102015

**Published:** 2023-09-27

**Authors:** Yueh Lin, Wan-Ying Lin, Ting-Wei Lin, Yi-Ju Tseng, Yu-Chiang Wang, Jia-Ruei Yu, Chia-Ru Chung, Hsin-Yao Wang

**Affiliations:** 1Department of Family Medicine, Chang Gung Memorial Hospital, Linkou Branch, Taoyuan 333, Taiwan; b0105032@cgmh.org.tw; 2Department of Medicine, University of California San Diego, San Diego, CA 92093, USA; wal018@health.ucsd.edu; 3Department of Medicine, Harvard Medical School, Boston, MA 02115, USA; ywang134@bwh.harvard.edu; 4Department of Medicine, Brigham and Women’s Hospital, Boston, MA 02115, USA; 5Department of Laboratory Medicine, Chang Gung Memorial Hospital, Linkou Branch, Taoyuan 333, Taiwan; weitinglin66@gmail.com (T.-W.L.); st33351995@gmail.com (J.-R.Y.); 6Department of Computer Science, National Yang Ming Chiao Tung University, Hsinchu 30010, Taiwan; yjtseng@nycu.edu.tw; 7Computational Health Informatics Program, Boston Children’s Hospital, Boston, MA 02115, USA; 8Department of Computer Science and Information Engineering, National Central University, Taoyuan 320317, Taiwan; jjrchris@g.ncu.edu.tw

**Keywords:** HPV, molecular epidemiology, prophylactic HPV vaccination, genotype distribution, high-risk genotypes

## Abstract

Cervical cancer, a major health concern among women worldwide, is closely linked to human papillomavirus (HPV) infection. This study explores the evolving landscape of HPV molecular epidemiology in Taiwan over a decade (2010–2020), where prophylactic HPV vaccination has been implemented since 2007. Analyzing data from 40,561 vaginal swab samples, with 42.0% testing positive for HPV, we reveal shifting trends in HPV genotype distribution and infection patterns. The 12 high-risk genotypes, in order of decreasing percentage, were HPV 52, 58, 16, 18, 51, 56, 39, 59, 33, 31, 45, and 35. The predominant genotypes were HPV 52, 58, and 16, accounting for over 70% of cases annually. The proportions of high-risk and non-high-risk HPV infections varied across age groups. High-risk infections predominated in sexually active individuals aged 30-50 and were mixed-type infections. The composition of high-risk HPV genotypes was generally stable over time; however, HPV31, 33, 39, and 51 significantly decreased over the decade. Of the strains, HPV31 and 33 are shielded by the nonavalent HPV vaccine. However, no reduction was noted for the other seven genotypes. This study offers valuable insights into the post-vaccine HPV epidemiology. Future investigations should delve into HPV vaccines’ effects and their implications for cervical cancer prevention strategies. These findings underscore the need for continued surveillance and research to guide effective public health interventions targeting HPV-associated diseases.

## 1. Introduction

Cervical cancer is one of the most common cancers threatening women’s health, and in recent decades, it has taken fourth place in both incidence and mortality among female cancers worldwide [1]. Human papillomavirus (HPV) is highly associated with cervical cancer in molecular and epidemiological studies [2,3,4,5]. According to the International Agency for Research on Cancer, 12 HPV genotypes, HPV16, 18, 31, 33, 35, 45, 52, 58, 39, 51, 56, and 59, are considered carcinogenic, referred to as high-risk types [6]. Regarding the molecular epidemiologies from 1990 to 2010, a meta-analysis including multiple studies across many countries showed that the most prevalent genotypes were HPV16, 18, 58, 33, 45, 31, 52, 35, 59, 39, 51, and 56 [7]. In Taiwan, the prevalence of HPV genotypes in cervical cancer was also reported in a study published in 2007 [8]. In the following decade, however, the molecular epidemiology of HPV in Taiwan was not reported. Prophylactic HPV vaccination has been widely used worldwide since 2006, and its impact on the circulating HPV strains is of interest.

The quadrivalent prophylactic HPV vaccine targeted against genotypes 6, 11, 16, and 18 was first licensed in 2006, the bivalent vaccine for genotypes 16 and 18 in 2007, and the nonavalent for genotypes 6, 11, 16, 18, 31, 33, 45, 52, and 58 was approved in 2014 [9]. All three vaccines against high-risk HPV types are composed of virus-like particles of the L1 protein, which is the major structural protein of the capsid of HPV, and are non-infectious due to the lack of viral DNA. HPV vaccination prevents new infection by the induction of antibodies that neutralize the infection [10,11,12]. HPV vaccines protect against genital warts and precancerous cervical lesions, as well against invasive cervical cancers [13,14,15,16]. As of 2020, more than 100 countries and territories had introduced HPV vaccination into national immunization programs [17,18]. In Taiwan, HPV vaccination was approved in 2007. The trends of the major circulating HPV strains are suspected to have shifted due to prophylactic vaccination since its introduction.

Knowledge about the distribution of population-based HPV genotypes in cervical cancer is crucial for developing HPV-based screening tests and evaluating the effects of prophylactic HPV vaccines. However, analyses of HPV genotype distribution following the implementation of prophylactic vaccination are lacking. Thus, we aimed to update the previous HPV-type-specific prevalence data by investigating the changes in prevalence during the 10 years of implementation of prophylactic HPV vaccines.

## 2. Material and Methods

### 2.1. Study Design and Data Source

In this retrospective study, patients were included from Linkou Chang Gung Memorial Hospital (CGMH) between 1 January 2010 and 31 December 2020. The CGMH is a tertiary referral medical center with more than 3000 acute care beds. Medical reimbursement from CGMH accounts for roughly 10% of the national healthcare expenditure. Samples included patients who had undergone HPV tests from out-patient departments or in-patient wards. Specimens were taken from cervical cells and investigated for their HPV genotype by PCR and gene chip hybridization. A high-risk-type infection is defined as detection of at least one of the twelve high-risk HPV genotypes, regardless of the presence of any other non-high-risk HPV genotype. The study protocol was approved by the Institutional Review Board of Chang Gung Memorial Hospital (202300572B1, approval date: 17 April 2023).

### 2.2. HPV DNA Extraction, PCR Amplification, and Genotyping

Specimens of cervical cells were tested at the central laboratory of CGMH. HPV DNA was extracted using the QIAamp DNA Blood Mini Kit (QIAGEN, Hilden, Germany) according to the manufacturer’s instructions. The PCR reaction was performed at a final volume of 25 µL, including PCR mix, primers, HotStart Taq DNA Polymerase, and DNA in PCR tubes of 0.1 mL. The assay included the amplification of a 136 bp fragment of glyceraldehyde-3-phosphate dehydrogenase (GAPDH) (internal control) and a 220 bp fragment of the HPV L1 region (MY11/GP6+). The resultant amplimers were then hybridized with an Easychip^®^ HPV Blot (King Car, I-Lan, Taiwan) (hereafter, HPV Blot) membrane. Thirty-eight types of HPV (6, 11, 16, 18, 26, 31, 32, 33, 35, 37, 39, 42, 43, 44, 45, 51, 52, 53, 54, 55, 56, 58, 59, 61, 62, 66, 67, 68, 69, 70, 71 (CP8061), 72, 74, 81 (CP8304), 82 (MM4), 83 (MM7), 84 (MM8), and L1AE5) could be identified.

### 2.3. Statistical Analysis

To facilitate our analyses, we defined a ‘single-type infection’ as an HPV infection where only one HPV genotype was identified in a sample and a ‘mixed-type infection’ as an HPV infection where multiple HPV genotypes were identified. HPV genotypes were grouped into high-risk and non-high-risk categories for subsequent analyses. Trend analyses were performed for the high-risk and non-high-risk HPV infection groups over 10 years and for individual HPV genotypes. In addition, we evaluated the trends of both high-risk and non-high-risk HPV infections across different age groups. The chi-squared test for trends in proportions was used in this study. This test allowed us to assess whether there was a trend in the proportions of high-risk and non-high-risk HPV infections over time. The null hypothesis was set as no trend in the proportions, while the alternative hypothesis was an increase or decrease in trend proportions. In addition to analyzing the overall trends in high-risk and non-high-risk HPV infections, the trends in individual HPV genotypes were evaluated. Trends in high-risk and non-high-risk HPV infections in different age groups were also examined. This information is valuable for the development of targeted prevention and screening strategies. All our statistical analyses were performed using R (version 4.2.2), a powerful programming language widely used in the scientific community for data analysis. Our findings may help to elucidate the epidemiology of HPV infection and inform public health policies aimed at reducing the burden of HPV-associated disease. 

## 3. Results

### 3.1. General Descriptive Statistics of HPV in 10 Years

In total, 40,561 cases from either the out-patient department or the in-patient ward between 1 January 2010, and 31 December 2020, were included in our study. Twenty male cases were excluded. The general HPV genotype distribution is shown in Table 1. There were 23,513 HPV-negative and 17,048 HPV-positive cases, among which 65.2% were single-type infections and 34.8% were mixed-type infections. The leading 10 types were HPV 52, HPV 58, HPV 16, HPV 53, HPV 51, HPV 62, HPV 18, HPV 56, HPV 70, and HPV 54. Among the leading 10 types, HPV 52, 58, 16, and 18 were more frequently associated with single-type infections (> 50%) than mixed-type infections.

### 3.2. Distribution of High-Risk and Non-High-Risk HPV Groups

HPV-positive patients were divided into two groups: the high-risk group, composed of 12 HPV types (HPV-16, 18, 31, 33, 35, 39, 45, 51, 52, 56, 58, and 59), and the non-high-risk group (HPV genotypes except the 12 HPV types). Generally, the proportion between high-risk and non-high-risk groups was similar over the 10 years (Figure 1). The high-risk group accounted for a higher percentage than the non-high-risk group. Specifically, the high-risk group accounted for the highest percentage in 2010 (79.3%) and the lowest in 2014 (61.7%). The percentage of high-risk groups clearly decreased from 2010 to 2014 (79.3% to 61.7%, *p*-value < 0.05). After 2014, the composition was stable, in which the high-risk group accounted for around 65% cases.

### 3.3. HPV Infection in Different Age Groups

The number of HPV-positive patients which were divided into high-risk and non-high-risk groups is shown by different age groups in Figure 2. The distribution pattern differed in different age groups. The percentage of high-risk HPV was higher in the 20–30 age group than non-high-risk genotypes, which was significantly higher in the 30–40 age group and was significantly low in the 50–60 age group (Appendix A). In both the high-risk and non-high-risk groups, a significant increase was noted in the 60–70 age group that increased with age (Appendix A). The difference between the two groups was marginal in the age groups of 40–50, 60–70, and >70. High-risk HPV infection was more frequently detected in the 30–50 age group. We also tested the change in trends over time for both high-risk and non-high-risk genotypes in the age groups. The results disclosed that cases with non-high-risk genotypes increased in the age groups of 40–50, 50–60, and 60–70 (Appendix A). In contrast, the 60–70 age group is the only group where high-risk genotypes increased over time (Appendix A). Moreover, both high-risk and non-high-risk genotypes increased over time in the 60–70 age group.

### 3.4. Single-Type and Mixed-Type HPV Infection in Different Age Groups

We further investigated the number of single-type and mixed-type HPV infections in different age groups (Figure 3). The number of single-type HPV infections was 996 in 2010 and 959 in 2020, on average 1009.3 patients per year. Compared to the single-type infection, 579 patients had mixed-type HPV infections in 2010 and there were 543 cases in 2020, on average 553.5 patients per year. Among single-type HPV infections (Figure 3A), the average numbers of high-risk-type infections over time in each age group (20–30, 30–40, 40–50, 50–60, 60–70, and >70) were 25.7, 135.7, 136.3, 89.5, 53.3, and 19.5. The numbers of non-high-risk genotype infections were 29.2, 118.3, 161.0, 140.3, 76.1, and 24.5 per year in each age group. In mixed-type infections (Figure 3B), the average numbers of high-risk-type infections over time in each age group were 31.5, 92.3, 73.4, 62.9, 43.4, and 17.4. These numbers for the non-high-risk genotypes were 16.2, 48.5, 50.1, 51.0, 33.7, and 13.4 per year in each age group.

In brief, for single-type infections, the numbers of patients with non-high-risk HPV genotype infections were generally higher than those with high-risk HPV genotypes (Figure 3A). In contrast, for mixed-type infections, the numbers of patients with high-risk HPV genotype infections were higher than those with non-high-risk genotypes (Figure 3B).

### 3.5. Distribution of the 12 High-Risk Genotypes

The proportion of the 12 high-risk genotypes in HPV-positive patients over the decade is shown in the stacked bar chart (Figure 4). The 12 high-risk genotypes, in order of decreasing percentage, were HPV 52, 58, 16, 18, 51, 56, 39, 59, 33, 31, 45, and 35. Among the 12 genotypes, seven genotypes, HPV 16, 18, 31, 33, 45, 52, and 58, were related to the nonavalent HPV vaccine. The top three genotypes (HPV 52, 58, and 16) accounted for more than 70 percent yearly; the lowest genotype was HPV 35. The ranking of genotypes did not change over the 10 years.

### 3.6. Change in Trends for the 12 High-Risk Genotypes

When the overall composition and ranking were stable, we further analyzed the change in trends for specific genotypes (Figure 5). The trends of HPV 31, 33, 39, and 51 significantly decreased and those of HPV 58 and 59 increased (Appendix A). There were no statistically significant changes for the other six genotypes, although HPV 16, 18, 35, and 45 decreased and HPV 52 and 56 were elevated (Appendix A). HPV 52 was the predominant high-risk type every year.

## 4. Discussion

The molecular epidemiology of HPV in recent decades in Taiwan was investigated to update the molecular epidemiology of HPV and reveal different HPV genotype rankings. Investigation of the distribution of 12 HPV high-risk types disclosed that the overall ratio of high-risk HPV versus non-high-risk HPV has not changed much from 2010 to 2020. However, we found that some high-risk HPV genotypes (i.e., genotypes 31, 33, 39, and 51) decreased over 10 years. The results provide insights into the change in HPV’s molecular epidemiology after the implementation of vaccination.

### 4.1. HPV Prevalence and Genotype Distribution

In the study, the positive rate of HPV infection was 42.0% (17,048/40,561). The leading 10 types were HPV 52, 58, 16, 53, 51, 62, 18, 56, 70, and 54. Amid the ten prevalent types, six were related to high-risk types of HPV (52, 58, 16, 51, 18, and 56). A previous study in Taiwan reported that the leading eight HPV genotypes in cervical cancer were HPV 16, 18, 58, 33, 52, 39, 45, and 31 [8]. However, the study focused on patients with cervical cancer only. However, the overall distribution of HPV genotypes is unknown. In contrast, this study included the general population, not only women with different cervical abnormalities. In addition, the HPV prevalence worldwide among women with normal cytological findings is estimated at around 12% [19], whereas in Asia it is around 14% [20]. A population-based study in China reported an HPV prevalence varying from 8.92% to 44.5% in different regions [21]. Our results showed a higher HPV prevalence at 42.03%. This high molecular prevalence may be associated with the specific geographic area. CGMH is the largest healthcare system in Taiwan, accounting for around 10% of national healthcare costs. Thus, the features of long-term surveillance and the sample size render the study representative enough for the geographic area and for the period of time.

### 4.2. High Risk Ratio in Different Age Group

The overall distribution of HPV infections showed an increase in the prevalence of high-risk compared to non-high-risk HPV genotypes. The high risk versus non-high risk case numbers were generally comparable in all the age groups, but a higher number was noted in the 30–40 age group. The global age-specific HPV prevalence is higher at younger ages [9]. The HPV infection rate was higher in women aged 30–50 in this study. We found a relatively high prevalence of high-risk HPV genotype infections in the age group of 20–40. Compared to the other age groups, such as 50–60 and 60–70, the age group of 20–40 is considered to be more sexually active. These results concur with reports of a similar trend [22,23]. However, in Southeast Asia, the prevalence of HPV infections in all age groups is low [24]. We also found the HPV infection rate was significantly increased in the 60–70 age group, which is consistent with a trend toward a high prevalence in older women due to reduced immune responses and hormonal changes [21,25,26].

### 4.3. Single- and Mixed-Type Infections in Different Age Groups

Based on a comparable number of high-risk and non-high-risk HPV strains, we conducted further individual-level analyses categorizing single- or mixed-type HPV infections. Interestingly, the high-risk versus non-high-risk pattern for single-type infections was distinct from mixed-type infections. For individuals with single-type HPV infections, non-high-risk genotypes were generally more prevalent than high-risk genotypes in all the age groups over time except the 30–40 age group. By contrast, there were more patients with a high-risk mixed-type infection than those with a non-high-risk mixed-type infection in all the age groups. Carcinogenesis in the high-risk HPV genotypes has been widely reported [6,27]. Multiple HPV infections are correlated with abnormal cervical lesions [28,29,30], and mixed-type infections with high-risk HPV strains are a key risk factor in the incidence of cervical carcinoma [31].

### 4.4. The Changing Trends of the 12 High-Risk Types

Over the decade, our study showed the overall proportion of high-risk HPV types reduced from 79.3% (2010) to 65.6% (2020). However, there was no change in the overall ranking of the top 12 high-risk HPV genotypes over time. HPV genotypes 52, 58, and 16 account for approximately more than 70 percent of the top 12 high-risk genotypes in accordance with previous studies in Asia and China [32,33]. Based on these data, this study focused on analyzing the changes in trends of the 12 high-risk genotypes and their prevalence. The genotypes HPV 52 and 58 were the most common high-risk genotypes in the study, although HPV types 16 and 18 are well known as the most frequent types worldwide and are strongly related to cervical cancer [9]. In addition, HPV 51, although not a common genotype worldwide, ranked fourth/fifth in the high-risk HPV types. The nonavalent HPV vaccine is ineffective against the HPV 51 genotype; thus, it requires special consideration when developing second-generation HPV prophylactic vaccines in the future.

The composition of high-risk HPV genotypes was generally stable over time; however, HPV 31, 33, 39, and 51 significantly decreased over time. Among the 12 genotypes, seven genotypes, HPV 16, 18, 31, 33, 45, 52, and 58, are shielded by the nonavalent HPV vaccine (excluding HPV 6 and 11). Prophylactic HPV vaccines can decrease cervical cancer rates [34]. After the implementation of a national immunization program, reductions in HPV vaccine-type prevalence and disease were noted in previous studies [35,36]. The reduction in HPV 31 and 33 could result from nonavalent vaccination, while other high-risk genotypes did not decrease between 2010 and 2020. A possible explanation could be that the nationwide HPV vaccination program of Taiwan started in 2018. Partial effects on reducing high-risk genotypes, including type 31 and 33, have been noted. Although there were no statistically significant changes, decreasing trends in HPV 16, 18, and 45 were also observed. The reduction in other high-risk genotypes may take time to record. HPV vaccines can reduce the prevalence of high-risk types, boosting vaccine implementation confidence. However, while some high-risk variants decrease, the prevalence of non-high-risk types, such as cutaneous beta HPV types associated with skin cancer, may increase [37]. Similar transition dynamics are found in the SARS-CoV-2 virus [38]. Thus, the strategy of developing novel HPV vaccines in the future should include not only traditional high-risk variants but also other disease-related HPV variants to have a more comprehensive protection against HPV-related diseases.

The importance of HPV vaccination for protection against cervical cancer and HPV-related diseases has been well established. However, not all countries have included HPV vaccines in their national immunization programs. Despite accounting for most of the disease burden in lower- and middle-income countries (LMICs), <30% of these nations have introduced the vaccine compared to >80% of high-income countries (HICs) [1,18]. The adoption rate remains vulnerable to multiple factors such as lack of government funding and political support, lack of public hygiene awareness, the absence of effective screening, employment levels, and national income [39,40]. These barriers might perpetuate HPV infection transmission, warranting consideration of formulating effective public health interventions. Thus, a continuous effort to broaden HPV vaccination is still needed.

### 4.5. Limitations

The limitations of the study include the lack of demographic information like sexual activity and history. There are more factors besides vaccination that could have an impact on the molecular epidemiology of HPV. This would limit the ability to establish causal relationships. Additionally, the study only included data from a single healthcare system in Taiwan. Although this healthcare system is the largest in Taiwan, the data may not represent the entire country or other regions. In addition, throughout the coronavirus disease 2019 (COVID-19) pandemic, COVID-19 has had a negative impact on the diagnosis of new HPV cases [41]. The study’s timeframe partially overlapped with the COVID-19 pandemic (December 2020). While this period accounts for a relatively small portion of the overall ten-year dataset, we acknowledge the potential impact of COVID in hindering the diagnosis of new HPV cases. Finally, this study focused on the prevalence of HPV infections and did not investigate the incidence of cervical cancer or other HPV-related diseases. The study provides a rough overview of HPV genotypes in a post-HPV vaccine era. Further study is needed to investigate the effects of the HPV vaccine.

## 5. Conclusions

We studied the molecular epidemiology of HPV in Taiwan for a decade (2010–2020). High-risk HPV genotypes were prevalent, especially in sexually active age groups and mixed-type infections. HPV 31 and 33, which are covered by the nonavalent HPV vaccine, decreased over time. However, reductions were not noted in the other seven genotypes.

## Figures and Tables

**Figure 1 viruses-15-02015-f001:**
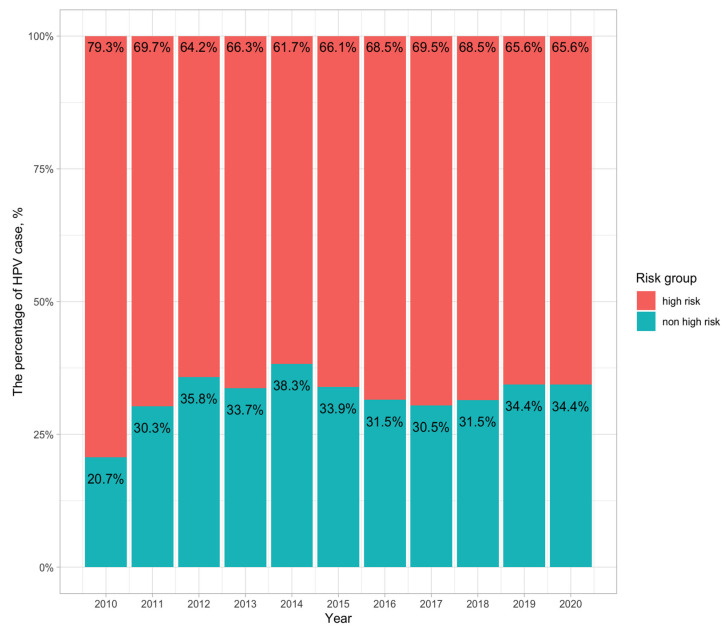
Distribution of high-risk and non-high-risk HPV groups from 2010 to 2020. Twelve HPV genotypes, including HPV-16, 18, 31, 33, 35, 39, 45, 51, 52, 56, 58, and 59, are referred to as high-risk types. HPV genotypes other than these 12 types are considered as non-high-risk types.

**Figure 2 viruses-15-02015-f002:**
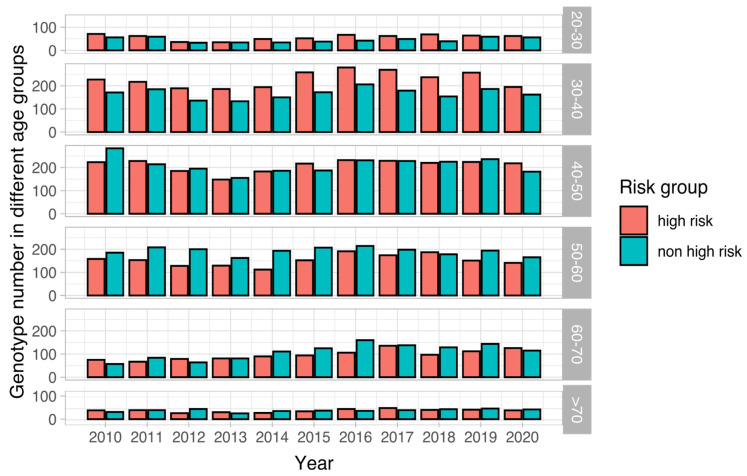
Distribution of high-risk and non-high-risk HPV genotypes in different age groups over 10 years. A biphasic distribution is noted for different age groups. For the age groups before age 50 (i.e., 20–30, 30–40, and 40–50), high-risk genotypes are more frequently detected than non-high-risk genotypes. In contrast, non-high-risk genotypes are more commonly detected than high-risk genotypes for the older age groups (i.e., >50 years old). Generally, the compositions did not change in the 10-year period.

**Figure 3 viruses-15-02015-f003:**
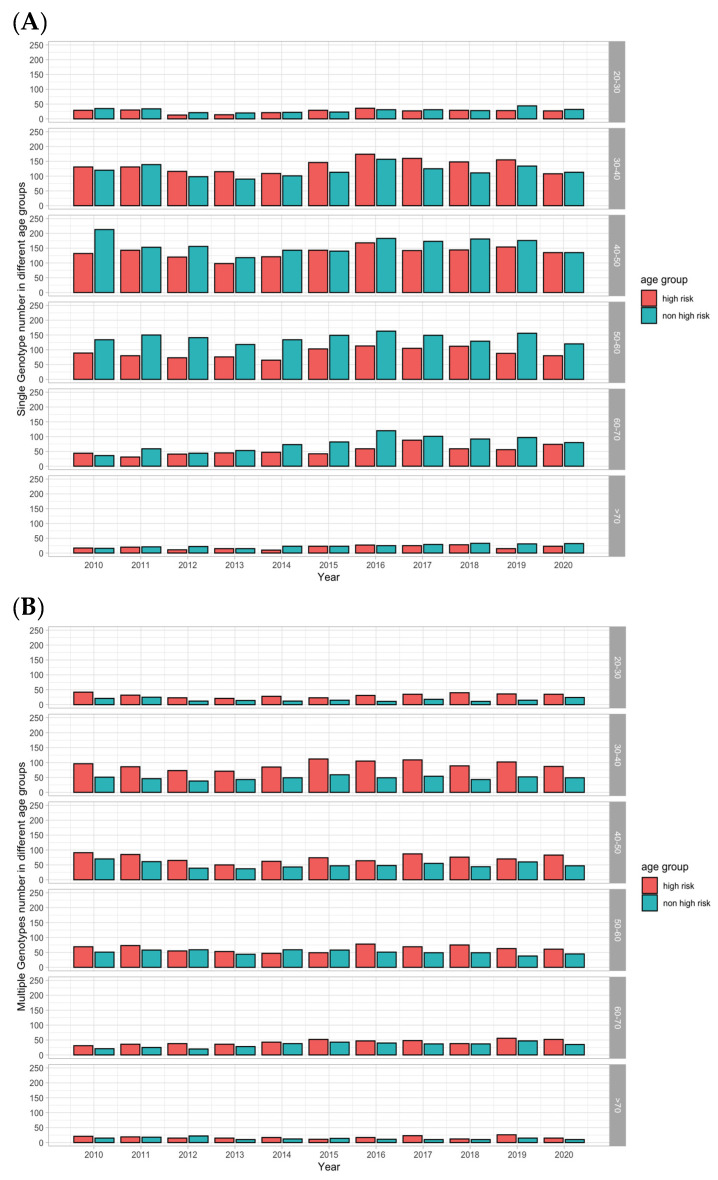
Single-type HPV and mixed-type HPV infections over time. (**A**) Single-type HPV infections. The number of non-high-risk HPV infections is generally larger than that of high-risk infections over time in each age group. (**B**) Mixed-type HPV infections. The number of high-risk HPV infections is larger than that of non-high-risk infections over time in each age group.

**Figure 4 viruses-15-02015-f004:**
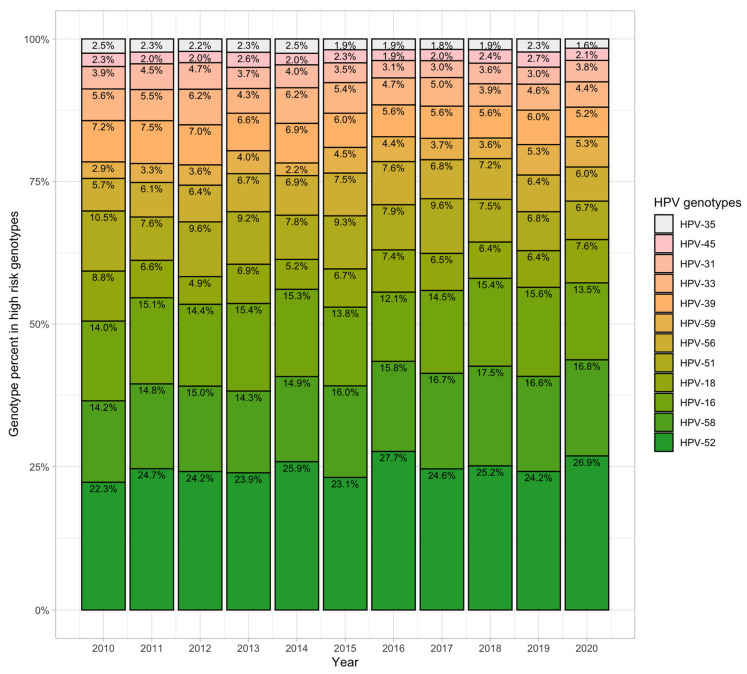
Distribution of the 12 high-risk HPV genotypes in the decade. The 12 high-risk HPV genotypes, in descending order, are HPV 52, 58, 16, 18, 51, 56, 39, 59, 33, 31, 45, and 35. Generally, the overall composition and ranking were stable over 10 years.

**Figure 5 viruses-15-02015-f005:**
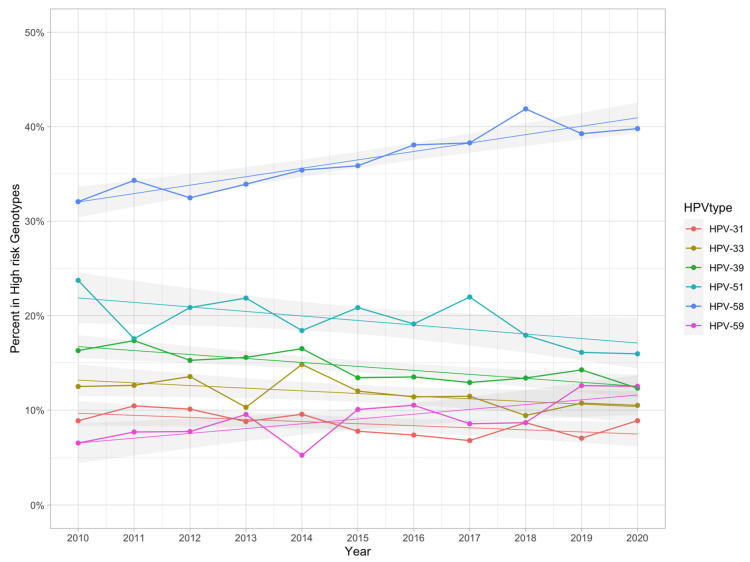
Change in trends for the 12 high-risk HPV genotypes. HPV 31, 33, 39, and 51 decreased significantly, while HPV 58 and 59 increased over 10 years. There were no statistically significant differences in the other six genotypes.

**Table 1 viruses-15-02015-t001:** Distribution of the HPV genotypes. * indicates high-risk genotypes.

HPV Genotype Distribution (N = 40,561)			
	No. of Patients (%)	Mixed-Type Infection	Single-Type Infection
Total	40,561 (100.0)		
HPV(−)	23,513 (58.0)	NA	NA
HPV(+)	17,048 (42.0)	34.8% (5932/17,048)	65.2% (11,116/17,048)
HPV-52*	3260 (8.0)	44.1% (1439/3260)	55.9% (1821/3260)
HPV-58*	2074 (5.1)	46.9% (972/2074)	53.1% (1102/2074)
HPV-16*	1894 (4.7)	44.3% (839/1894)	55.7% (1055/1894)
HPV-53	1622 (4.0)	58.8% (954/1622)	41.2% (668/1622)
HPV-51*	1106 (2.7)	57.6% (637/1106)	42.4% (469/1106)
HPV-62	1022 (2.5)	63.8% (652/1022)	36.0% (368/1022)
HPV-18*	884 (2.2)	46.4% (410/884)	53.6% (474/884)
HPV-56*	878 (2.2)	60.9% (535/878)	39.1% (343/878)
HPV-70	859 (2.1)	56.0% (481/859)	44.0% (378/859)
HPV-54	850 (2.1)	70.8% (602/850)	29.2% (248/850)
HPV-39*	823 (2.0)	61.1% (503/823)	38.9% (320/823)
HPV-42	814 (2.0)	61.5% (501/814)	38.5% (313/814)
HPV-84(MM8)	737 (1.8)	70.6% (520/737)	29.4% (217/737)
HPV-81(CP8304)	716 (1.8)	67.0% (480/716)	33.0% (236/716)
HPV-66	705 (1.7)	63.0% (444/705)	37.0% (261/705)
HPV-33*	664 (1.6)	52.9% (351/664)	47.1% (313/664)
HPV-61	627 (1.5)	68.1% (427/627)	31.9% (200/627)
HPV-44	580 (1.4)	66.9% (388/580)	33.1% (192/580)
HPV-72	554 (1.4)	57.9% (321/554)	42.1% (233/554)
HPV-68	548 (1.4)	60.9% (334/548)	38.5% (211/548)
HPV-59*	517 (1.3)	66.9% (346/517)	33.1% (171/517)
HPV-71(CP8061)	502 (1.2)	62.5% (314/502)	37.5% (188/502)
HPV-31*	480 (1.2)	58.3% (280/480)	41.7% (200/480)
HPV-43	476 (1.2)	67.9% (323/476)	32.1% (153/476)
HPV-82(MM4)	415 (1.0)	77.3% (321/415)	22.7% (94/415)
HPV-6	410 (1.0)	66.8% (274/410)	33.2% (136/410)
HPV-45*	290 (0.7)	66.6% (193/290)	33.4% (97/290)
HPV-35*	275 (0.7)	58.2% (160/275)	41.8% (115/275)
HPV-67	265 (0.7)	75.1% (199/265)	24.9% (66/265)
HPV-55	257 (0.6)	75.1% (193/257)	24.9% (64/257)
HPV-74	248 (0.6)	54.0% (134/248)	46.0% (114/248)
HPV-69	199 (0.5)	65.8% (131/199)	34.2% (68/199)
HPV-11	193 (0.5)	49.2% (95/193)	50.8% (98/193)
HPV-32	164 (0.4)	73.8% (121/164)	26.2% (43/164)
HPV-L1AE5	95 (0.2)	73.7% (70/95)	26.3% (25/95)
HPV-83(MM7)	82 (0.2)	42.7% (35/82)	57.3% (47/82)
HPV-26	64 (0.2)	76.6% (49/64)	23.4% (15/64)
HPV-37	2 (0.0)	100.0% (2/2)	0.0% (0/2)

## Data Availability

The data presented in this study are available on request from the corresponding author. The data are not publicly available due to privacy.

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
