# Peer review of "Trend of HPV Molecular Epidemiology in the Post-Vaccine Era: A 10-Year Study"

_viruses, 2023, doi:10.3390/v15102015_

Round 1
Reviewer 1 Report
This study examines the evolution of HPV molecular epidemiology in Taiwan following prophylactic HPV vaccination. The analysis period is 10 years. In the analysis, the occurrence of high-risk and non-high-risk HPV genotypes are recorded over a 10-year analysis period. The study is comprehensive, well-structured, and provides sound information on HPV epidemiology after vaccination. The identified changes in individual high-risk genotypes (31 and 33) are discussed in the context of the national HPV vaccination program, underscoring the need for and success of a national HPV vaccine.
From my side, the submitted manuscript is more published in its current form.
However, I suggest that in the results section 3.1. the indication of percentage values in the text could be omitted, since these are already listed in the adjacent table. This would facilitate the readability of the passage. Also, the specification of figures in the Discussion section is unusual and not necessary, since it distracts from the actual discussion and looks like the Result section.
Author Response
Responses to Reviewer #1:
This study examines the evolution of HPV molecular epidemiology in Taiwan following prophylactic HPV vaccination. The analysis period is 10 years. In the analysis, the occurrence of high-risk and non-high-risk HPV genotypes are recorded over a 10-year analysis period. The study is comprehensive, well-structured, and provides sound information on HPV epidemiology after vaccination. The identified changes in individual high-risk genotypes (31 and 33) are discussed in the context of the national HPV vaccination program, underscoring the need for and success of a national HPV vaccine.
From my side, the submitted manuscript is more published in its current form. However, I suggest that in the results section 3.1. the indication of percentage values in the text could be omitted, since these are already listed in the adjacent table. This would facilitate the readability of the passage. Also, the specification of figures in the Discussion section is unusual and not necessary, since it distracts from the actual discussion and looks like the Result section.
Response:
Based on your nice suggestion, we have removed the percentage values in the text. Moreover, the specification of figures is also removed to enhance the readability. Please refer to the modifications to page 3,9,10.
Reviewer 2 Report
Lin and colleagues conducted a comprehensive investigation into the evolving patterns of HPV infection in Taiwan spanning a decade (2010-2020), synchronizing with the introduction of HPV vaccination. By meticulously analyzing data from 40,561 samples, the study unveils noteworthy shifts in prevalent HPV genotypes. Particularly striking is the prominence of HPV 52, 58, and 16, collectively constituting more than 70% of reported cases. Delving deeper into the implications of these shifting trends in HPV genotype distribution could provide valuable insights into the driving forces behind these changes, potentially shedding light on aspects like virulence and transmission dynamics. Moreover, considering the observed variation in high-risk and non-high-risk HPV infections across diverse age groups, the paper could offer an extended discussion on the potential reasons underlying this pattern, elucidating the potential role of sexual behavior, immune response, and overall HPV persistence. Notably, the heightened prevalence of high-risk infections within the sexually active 30-50 age range might have implications for targeted vaccination strategies, warranting a more detailed exploration.
While the study effectively addresses the reduction of HPV 31 and 33, facilitated by their coverage in the nonavalent HPV vaccine, a deeper investigation into the stability of the other seven genotypes could further strengthen the paper's conclusions. Furthermore, given the relevance of beta HPV types in causing skin cancer, particularly in immunocompromised individuals and those with epidermodysplasia verruciformis, the researchers could consider discussing the broader ramifications of their findings on existing vaccination strategies. Exploring potential correlations between the study's outcomes and beta HPV type prevalence may provide a valuable avenue for future research in this domain.
Regarding the impact of the COVID-19 pandemic on HPV infection rates, it would be prudent to incorporate a brief discussion, especially since some of the analyzed samples pertain to the years 2019-2020. This would not only enhance the paper's relevance but also acknowledge the potential influence of external factors on the study's findings. Citing the paper with DOI 10.3389/fpubh.2022.880435 could aptly support this discussion.
In the context of global HPV vaccination efforts, particularly in low- and middle-income countries, highlighting the barriers to vaccine uptake identified in studies with DOIs 10.3389/fimmu.2023.1150238 and 10.1111/ajco.13513 would fortify the paper's real-world implications. These barriers might perpetuate HPV infection transmission, warranting consideration in the formulation of effective public health interventions. In conclusion, while the study offers intriguing insights, addressing these suggestions could elevate the paper's impact and comprehensiveness.
Author Response
Responses to Reviewer #2:
Lin and colleagues conducted a comprehensive investigation into the evolving patterns of HPV infection in Taiwan spanning a decade (2010-2020), synchronizing with the introduction of HPV vaccination. By meticulously analyzing data from 40,561 samples, the study unveils noteworthy shifts in prevalent HPV genotypes. Particularly striking is the prominence of HPV 52, 58, and 16, collectively constituting more than 70% of reported cases. Delving deeper into the implications of these shifting trends in HPV genotype distribution could provide valuable insights into the driving forces behind these changes, potentially shedding light on aspects like virulence and transmission dynamics. Moreover, considering the observed variation in high-risk and non-high-risk HPV infections across diverse age groups, the paper could offer an extended discussion on the potential reasons underlying this pattern, elucidating the potential role of sexual behavior, immune response, and overall HPV persistence. Notably, the heightened prevalence of high-risk infections within the sexually active 30-50 age range might have implications for targeted vaccination strategies, warranting a more detailed exploration.
Response:
We appreciate the reviewer's kind words and recognition of the value of our study.
While the study effectively addresses the reduction of HPV 31 and 33, facilitated by their coverage in the nonavalent HPV vaccine, a deeper investigation into the stability of the other seven genotypes could further strengthen the paper's conclusions. Furthermore, given the relevance of beta HPV types in causing skin cancer, particularly in immunocompromised individuals and those with epidermodysplasia verruciformis, the researchers could consider discussing the broader ramifications of their findings on existing vaccination strategies. Exploring potential correlations between the study's outcomes and beta HPV type prevalence may provide a valuable avenue for future research in this domain.
Response:
Your insightful comments provide a broader view for our work. The reduction of HPV 31 and 33 could be resulted from nonavalent vaccination, while other high-risk genotypes did not decrease in 2010-2020. The possible explanation could be that the nationwide HPV vaccination program of Taiwan started in 2018. Partial effects on reducing high-risk genotypes, including type 31 and 33, have been noted. Although there were no statistically significant changes, decreasing trends in HPV 16, 18 and 45 have also been observed. The further reducing effects on other high-risk genotypes would take more time to record. HPV vaccines can reduce high-risk types prevalence, boosting the confidence in vaccine implementation. However, while some high-risk variants decrease, the prevalence of non-high risk types such as cutaneous beta HPV types associated with skin cancer may increase.(doi: 10.1128/JVI.01003-18) Similar transition dynamics is found in SARS-CoV-2 virus.(doi: 10.1016/j.ebiom.2023.104534) Thus, the strategy of developing novel HPV vaccines in the future would include not only traditional high-risk variants but also other diseases-related HPV variants to have a more comprehensive protection against HPV-related diseases.
Please refer to page 11 for the amended paragraph.
“The reduction of HPV 31 and 33 could be resulted from nonavalent vaccination, while other high-risk genotypes did not decrease in 2010-2020. The possible explanation could be that the nationwide HPV vaccination program of Taiwan started in 2018. Partial effects on reducing high-risk genotypes, including type 31 and 33, have been noted. Although there were no statistically significant changes, decreasing trends in HPV 16, 18 and 45 have also been observed. The further reducing effects on other high-risk genotypes would take more time to record. HPV vaccines can reduce high-risk types prevalence, boosting the confidence in vaccine implementation. However, while some high-risk variants decrease, the prevalence of non-high risk types such as cutaneous beta HPV types associated with skin cancer may increase.37 Similar transition dynamics is found in SARS-CoV-2 virus.38 Thus, the strategy of developing novel HPV vaccines in the future would include not only traditional high-risk variants but also other diseases-related HPV variants to have a more comprehensive protection against HPV-related diseases.”
Regarding the impact of the COVID-19 pandemic on HPV infection rates, it would be prudent to incorporate a brief discussion, especially since some of the analyzed samples pertain to the years 2019-2020. This would not only enhance the paper's relevance but also acknowledge the potential influence of external factors on the study's findings. Citing the paper with DOI 10.3389/fpubh.2022.880435 could aptly support this discussion.
Response:
Thank you for pointing out the possible limitation. Indeed, throughout the coronavirus disease 2019 (COVID-19) pandemic, COVID-19 has had a negative impact on the diagnosis of new HPV cases. (10.3389/fpubh.2022.880435) The study's timeframe partially overlapped with the COVID-19 pandemic (December 2020). While this period accounts for a relatively small portion of the overall ten-year dataset, we acknowledge the potential impact of COVID in hindering the diagnosis of new HPV cases.
Please refer to page 11 for the amended paragraph.
“Besides, throughout the coronavirus disease 2019 (COVID-19) pandemic, COVID-19 has had a negative impact on the diagnosis of new HPV cases.41 The study's timeframe partially overlapped with the COVID-19 pandemic (December 2020). While this period accounts for a relatively small portion of the overall ten-year dataset, we acknowledge the potential impact of COVID in hindering the diagnosis of new HPV cases.”
In the context of global HPV vaccination efforts, particularly in low- and middle-income countries, highlighting the barriers to vaccine uptake identified in studies with DOIs 10.3389/fimmu.2023.1150238 and 10.1111/ajco.13513 would fortify the paper's real-world implications. These barriers might perpetuate HPV infection transmission, warranting consideration in the formulation of effective public health interventions. In conclusion, while the study offers intriguing insights, addressing these suggestions could elevate the paper's impact and comprehensiveness.
Response:
The importance of HPV vaccination for the protection against cervical cancer and HPV-related diseases has been well established. However, not all countries have included HPV vaccines into their national immunization programs. Despite accounting for most of the disease burden in lower-middle-income countries (LMICs), <30% of these nations have presented the vaccine compared with >80% of high-income countries (HICs). 1, 18 The adoption rate remains vulnerable to multi-factors such as lack of funding and political support from the government, lack of awareness among public hygiene, the absence of effective screening, employment levels, and national income. (DOIs 10.3389/fimmu.2023.1150238 and 10.1111/ajco.13513) These barriers might perpetuate HPV infection transmission, warranting consideration in the formulation of effective public health interventions. Thus, a continuous effort on broadening HPV vaccination is still needed.
Please refer to page 11 for the amended paragraph.
“The importance of HPV vaccination for the protection against cervical cancer and HPV-related diseases has been well established. However, not all countries have included HPV vaccines into their national immunization programs. Despite accounting for most of the disease burden in lower-middle-income countries (LMICs), <30% of these nations have presented the vaccine compared with >80% of high-income countries (HICs). 1, 18 The adoption rate remains vulnerable to multi-factors such as lack of funding and political support from the government, lack of awareness among public hygiene, the absence of effective screening, employment levels, and national income.39, 40 These barriers might perpetuate HPV infection transmission, warranting consideration in the formulation of effective public health interventions. Thus, a continuous effort on broadening HPV vaccination is still needed.”